# Matching Networks for One Shot Learning

**Oriol Vinyals**
Google DeepMind
vinyals@google.com

**Charles Blundell**
Google DeepMind
cblundell@google.com

**Timothy Lillicrap**
Google DeepMind
countzero@google.com

**Koray Kavukcuoglu**
Google DeepMind
korayk@google.com

**Daan Wierstra**
Google DeepMind
wierstra@google.com

## Abstract

Learning from a few examples remains a key challenge in machine learning. Despite recent advances in important domains such as vision and language, the standard supervised deep learning paradigm does not offer a satisfactory solution for learning new concepts rapidly from little data. In this work, we employ ideas from metric learning based on deep neural features and from recent advances that augment neural networks with external memories. Our framework learns a network that maps a small labelled support set and an unlabelled example to its label, obviating the need for fine-tuning to adapt to new class types. We then define one-shot learning problems on vision (using Omniglot, ImageNet) and language tasks. Our algorithm improves one-shot accuracy on ImageNet from 87.6% to 93.2% and from 88.0% to 93.8% on Omniglot compared to competing approaches. We also demonstrate the usefulness of the same model on language modeling by introducing a one-shot task on the Penn Treebank.

## 1 Introduction

Humans learn new concepts with very little supervision – e.g. a child can generalize the concept of "giraffe" from a single picture in a book – yet our best deep learning systems need hundreds or thousands of examples. This motivates the setting we are interested in: "one-shot" learning, which consists of learning a class from a single labelled example.

Deep learning has made major advances in areas such as speech [7], vision [13] and language [16], but is notorious for requiring large datasets. Data augmentation and regularization techniques alleviate overfitting in low data regimes, but do not solve it. Furthermore, learning is still slow and based on large datasets, requiring many weight updates using stochastic gradient descent. This, in our view, is mostly due to the parametric aspect of the model, in which training examples need to be slowly learnt by the model into its parameters.

In contrast, many non-parametric models allow novel examples to be rapidly assimilated, whilst not suffering from catastrophic forgetting. Some models in this family (e.g., nearest neighbors) do not require any training but performance depends on the chosen metric [1]. Previous work on metric learning in non-parametric setups [18] has been influential on our model, and we aim to incorporate the best characteristics from both parametric and non-parametric models – namely, rapid acquisition of new examples while providing excellent generalisation from common examples.

The novelty of our work is twofold: at the modeling level, and at the training procedure. We propose Matching Nets, a neural network which uses recent advances in attention and memory that enable rapid learning. Secondly, our training procedure is based on a simple machine learning principle: test and train conditions must match. Thus to train our network to do rapid learning, we train it by

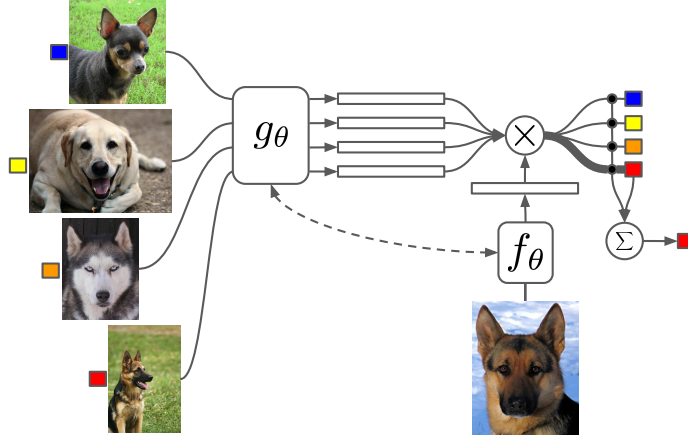

Figure 1: Matching Networks architecture

showing only a few examples per class, switching the task from minibatch to minibatch, much like how it will be tested when presented with a few examples of a new task.

Besides our contributions in defining a model and training criterion amenable for one-shot learning, we contribute by the definition of two new tasks that can be used to benchmark other approaches on both ImageNet and small scale language modeling. We hope that our results will encourage others to work on this challenging problem.

We organized the paper by first defining and explaining our model whilst linking its several components to related work. Then in the following section we briefly elaborate on some of the related work to the task and our model. In Section 4 we describe both our general setup and the experiments we performed, demonstrating strong results on one-shot learning on a variety of tasks and setups.

## 2  Model

Our non-parametric approach to solving one-shot learning is based on two components which we describe in the following subsections. First, our model architecture follows recent advances in neural networks augmented with memory (as discussed in Section 3). Given a (small) support set $S$, our model defines a function $c_S$ (or classifier) for each $S$, i.e. a mapping $S \rightarrow c_S(.)$. Second, we employ a training strategy which is tailored for one-shot learning from the support set $S$.

### 2.1  Model Architecture

In recent years, many groups have investigated ways to augment neural network architectures with external memories and other components that make them more "computer-like". We draw inspiration from models such as sequence to sequence (seq2seq) with attention [2], memory networks [29] and pointer networks [27].

In all these models, a neural attention mechanism, often fully differentiable, is defined to access (or read) a memory matrix which stores useful information to solve the task at hand. Typical uses of this include machine translation, speech recognition, or question answering. More generally, these architectures model $P(B|A)$ where $A$ and/or $B$ can be a sequence (like in seq2seq models), or, more interestingly for us, a set [26].

Our contribution is to cast the problem of one-shot learning within the set-to-set framework [26]. The key point is that when trained, Matching Networks are able to produce sensible test labels for unobserved classes *without any changes to the network*. More precisely, we wish to map from a (small) support set of $k$ examples of input-label pairs $S = \{(x_i, y_i)\}_{i=1}^{k}$ to a classifier $c_S(\hat{x})$ which, given a test example $\hat{x}$, defines a probability distribution over outputs $\hat{y}$. Here, $\hat{x}$ could be an image, and $\hat{y}$ a distribution over possible visual classes. We define the mapping $S \rightarrow c_S(\hat{x})$ to be $P(\hat{y}|\hat{x}, S)$ where $P$ is parameterised by a neural network. Thus, when given a new support set of examples $S'$ from which to one-shot learn, we simply use the parametric neural network defined by $P$ to make predictions about the appropriate label distribution $\hat{y}$ for each test example $\hat{x}$: $P(\hat{y}|\hat{x}, S')$.

Our model in its simplest form computes a probability over $\hat{y}$ as follows:

$$P(\hat{y}|\hat{x}, S) = \sum_{i=1}^{k} a(\hat{x}, x_i) y_i \qquad (1)$$

where $x_i, y_i$ are the inputs and corresponding label distributions from the support set $S = \{(x_i, y_i)\}_{i=1}^{k}$, and $a$ is an attention mechanism which we discuss below. Note that eq. 1 essentially describes the output for a new class as a linear combination of the labels in the support set. Where the attention mechanism $a$ is a kernel on $X \times X$, then (1) is akin to a kernel density estimator. Where the attention mechanism is zero for the $b$ furthest $x_i$ from $\hat{x}$ according to some distance metric and an appropriate constant otherwise, then (1) is equivalent to '$k - b$'-nearest neighbours (although this requires an extension to the attention mechanism that we describe in Section 2.1.2). Thus (1) subsumes both KDE and kNN methods. Another view of (1) is where $a$ acts as an attention mechanism and the $y_i$ act as values bound to the corresponding keys $x_i$, much like a hash table. In this case we can understand this as a particular kind of associative memory where, given an input, we "point" to the corresponding example in the support set, retrieving its label. Hence the functional form defined by the classifier $c_S(\hat{x})$ is very flexible and can adapt easily to any new support set.

### 2.1.1 The Attention Kernel

Equation 1 relies on choosing $a(.,.)$, the attention mechanism, which fully specifies the classifier. The simplest form that this takes (and which has very tight relationships with common attention models and kernel functions) is to use the softmax over the cosine distance $c$, i.e., $a(\hat{x}, x_i) = e^{c(f(\hat{x}), g(x_i))} / \sum_{j=1}^{k} e^{c(f(\hat{x}), g(x_j))}$ with embedding functions $f$ and $g$ being appropriate neural networks (potentially with $f = g$) to embed $\hat{x}$ and $x_i$. In our experiments we shall see examples where $f$ and $g$ are parameterised variously as deep convolutional networks for image tasks (as in VGG[22] or Inception[24]) or a simple form word embedding for language tasks (see Section 4).

We note that, though related to metric learning, the classifier defined by Equation 1 is discriminative. For a given support set $S$ and sample to classify $\hat{x}$, it is enough for $\hat{x}$ to be sufficiently aligned with pairs $(x', y') \in S$ such that $y' = y$ and misaligned with the rest. This kind of loss is also related to methods such as Neighborhood Component Analysis (NCA) [18], triplet loss [9] or large margin nearest neighbor [28].

However, the objective that we are trying to optimize is precisely aligned with multi-way, one-shot classification, and thus we expect it to perform better than its counterparts. Additionally, the loss is simple and differentiable so that one can find the optimal parameters in an "end-to-end" fashion.

### 2.1.2 Full Context Embeddings

The main novelty of our model lies in reinterpreting a well studied framework (neural networks with external memories) to do one-shot learning. Closely related to metric learning, the embedding functions $f$ and $g$ act as a lift to feature space $X$ to achieve maximum accuracy through the classification function described in eq. 1.

Despite the fact that the classification strategy is fully conditioned on the whole support set through $P(.|\hat{x}, S)$, the embeddings on which we apply the cosine similarity to "attend", "point" or simply compute the nearest neighbor are myopic in the sense that each element $x_i$ gets embedded by $g(x_i)$ independently of other elements in the support set $S$. Furthermore, $S$ should be able to modify how we embed the test image $\hat{x}$ through $f$.

We propose embedding the elements of the set through a function which takes as input the full set $S$ in addition to $x_i$, i.e. $g$ becomes $g(x_i, S)$. Thus, as a function of the whole support set $S$, $g$ can modify how to embed $x_i$. This could be useful when some element $x_j$ is very close to $x_i$, in which case it may be beneficial to change the function with which we embed $x_i$ – some evidence of this is discussed in Section 4. We use a bidirectional Long-Short Term Memory (LSTM) [8] to encode $x_i$ in the context of the support set $S$, considered as a sequence (see Appendix).

The second issue to make $f$ depend on $\hat{x}$ and $S$ can be fixed via an LSTM with read-attention over the whole set $S$, whose inputs are equal to $f'(\hat{x})$ ($f'$ is an embedding function, e.g. a CNN). To do

so, we define the following recurrence over "processing" steps $k$, following work from [26]:

$$\hat{h}_k, c_k = \text{LSTM}(f'(\hat{x}), [h_{k-1}, r_{k-1}], c_{k-1}) \quad (2)$$

$$h_k = \hat{h}_k + f'(\hat{x}) \quad (3)$$

$$r_{k-1} = \sum_{i=1}^{|S|} a(h_{k-1}, g(x_i))g(x_i) \quad (4)$$

$$a(h_{k-1}, g(x_i)) = e^{h_{k-1}^T g(x_i)} / \sum_{j=1}^{|S|} e^{h_{k-1}^T g(x_j)} \quad (5)$$

noting that $\text{LSTM}(x, h, c)$ follows the same LSTM implementation defined in [23] with $x$ the input, $h$ the output (i.e., cell after the output gate), and $c$ the cell. $a$ is commonly referred to as "content" based attention. We do $K$ steps of "reads", so $f(\hat{x}, S) = h_K$ where $h_k$ is as described in eq. 3.

## 2.2 Training Strategy

In the previous subsection we described Matching Networks which map a support set to a classification function, $S \rightarrow c(\hat{x})$. We achieve this via a modification of the set-to-set paradigm augmented with attention, with the resulting mapping being of the form $P_\theta(.|\hat{x}, S)$, noting that $\theta$ are the parameters of the model (i.e. of the embedding functions $f$ and $g$ described previously).

The training procedure has to be chosen carefully so as to match inference at test time. Our model has to perform well with support sets $S'$ which contain classes never seen during training.

More specifically, let us define a task $T$ as distribution over possible label sets $L$. Typically we consider $T$ to uniformly weight all data sets of up to a few unique classes (e.g., 5), with a few examples per class (e.g., up to 5). In this case, a label set $L$ sampled from a task $T$, $L \sim T$, will typically have 5 to 25 examples.

To form an "episode" to compute gradients and update our model, we first sample $L$ from $T$ (e.g., $L$ could be the label set $\{cats, dogs\}$). We then use $L$ to sample the support set $S$ and a batch $B$ (i.e., both $S$ and $B$ are labelled examples of cats and dogs). The Matching Net is then trained to minimise the error predicting the labels in the batch $B$ conditioned on the support set $S$. This is a form of meta-learning since the training procedure explicitly learns to learn from a given support set to minimise a loss over a batch. More precisely, the Matching Nets training objective is as follows:

$$\theta = \arg\max_\theta E_{L \sim T} \left[ E_{S \sim L, B \sim L} \left[ \sum_{(x,y) \in B} \log P_\theta(y|x, S) \right] \right]. \quad (6)$$

Training $\theta$ with eq. 6 yields a model which works well when sampling $S' \sim T'$ from a different distribution of novel labels. Crucially, our model does not need any fine tuning on the classes it has never seen due to its non-parametric nature. Obviously, as $T'$ diverges far from the $T$ from which we sampled to learn $\theta$, the model will not work – we belabor this point further in Section 4.1.2.

## 3 Related Work

### 3.1 Memory Augmented Neural Networks

A recent surge of models which go beyond "static" classification of fixed vectors onto their classes has reshaped current research and industrial applications alike. This is most notable in the massive adoption of LSTMs [8] in a variety of tasks such as speech [7], translation [23, 2] or learning programs [4, 27]. A key component which allowed for more expressive models was the introduction of "content" based attention in [2], and "computer-like" architectures such as the Neural Turing Machine [4] or Memory Networks [29]. Our work takes the metalearning paradigm of [21], where an LSTM learnt to learn quickly from data presented sequentially, but we treat the data as a set. The one-shot learning task we defined on the Penn Treebank [15] relates to evaluation techniques and models presented in [6], and we discuss this in Section 4.

## 3.2 Metric Learning

As discussed in Section 2, there are many links between content based attention, kernel based nearest neighbor and metric learning [1]. The most relevant work is Neighborhood Component Analysis (NCA) [18], and the follow up non-linear version [20]. The loss is very similar to ours, except we use the whole support set $S$ instead of pair-wise comparisons which is more amenable to one-shot learning. Follow-up work in the form of deep convolutional siamese [11] networks included much more powerful non-linear mappings. Other losses which include the notion of a set (but use less powerful metrics) were proposed in [28].

Lastly, the work in one-shot learning in [14] was inspirational and also provided us with the invaluable Omniglot dataset – referred to as the "transpose" of MNIST. Other works used zero-shot learning on ImageNet, e.g. [17]. However, there is not much one-shot literature on ImageNet, which we hope to amend via our benchmark and task definitions in the following section.

## 4 Experiments

In this section we describe the results of many experiments, comparing our Matching Networks model against strong baselines. All of our experiments revolve around the same basic task: an $N$-way $k$-shot learning task. Each method is providing with a set of $k$ labelled examples from each of $N$ classes that have not previously been trained upon. The task is then to classify a disjoint batch of unlabelled examples into one of these $N$ classes. Thus random performance on this task stands at $1/N$. We compared a number of alternative models, as baselines, to Matching Networks.

Let $L'$ denote the held-out subset of labels which we only use for one-shot. Unless otherwise specified, training is always on $\neq L'$, and test in one-shot mode on $L'$.

We ran one-shot experiments on three data sets: two image classification sets (Omniglot [14] and ImageNet [19, ILSVRC-2012]) and one language modeling (Penn Treebank). The experiments on the three data sets comprise a diverse set of qualities in terms of complexity, sizes, and modalities.

## 4.1 Image Classification Results

For vision problems, we considered four kinds of baselines: matching on raw pixels, matching on discriminative features from a state-of-the-art classifier (Baseline Classifier), MANN [21], and our reimplementation of the Convolutional Siamese Net [11]. The baseline classifier was trained to classify an image into one of the original classes present in the training data set, but excluding the $N$ classes so as not to give it an unfair advantage (i.e., trained to classify classes in $\neq L'$). We then took this network and used the features from the last layer (before the softmax) for nearest neighbour matching, a strategy commonly used in computer vision [3] which has achieved excellent results across many tasks. Following [11], the convolutional siamese nets were trained on a same-or-different task of the original training data set and then the last layer was used for nearest neighbour matching.

| Model | Matching Fn | Fine Tune | 5-way Acc 1-shot | 5-shot | 20-way Acc 1-shot | 5-shot |
|---|---|---|---|---|---|---|
| **PIXELS** | Cosine | N | 41.7% | 63.2% | 26.7% | 42.6% |
| **BASELINE CLASSIFIER** | Cosine | N | 80.0% | 95.0% | 69.5% | 89.1% |
| **BASELINE CLASSIFIER** | Cosine | Y | 82.3% | 98.4% | 70.6% | 92.0% |
| **BASELINE CLASSIFIER** | Softmax | Y | 86.0% | 97.6% | 72.9% | 92.3% |
| **MANN (NO CONV) [21]** | Cosine | N | 82.8% | 94.9% | – | – |
| **CONVOLUTIONAL SIAMESE NET [11]** | Cosine | N | 96.7% | 98.4% | 88.0% | 96.5% |
| **CONVOLUTIONAL SIAMESE NET [11]** | Cosine | Y | 97.3% | 98.4% | 88.1% | 97.0% |
| **MATCHING NETS (OURS)** | Cosine | N | **98.1%** | **98.9%** | **93.8%** | 98.5% |
| **MATCHING NETS (OURS)** | Cosine | Y | 97.9% | 98.7% | 93.5% | **98.7%** |

Table 1: Results on the Omniglot dataset.

We also tried further fine tuning the features using only the support set $S'$ sampled from $L'$. This yields massive overfitting, but given that our networks are highly regularized, can yield extra gains. Note that, even when fine tuning, the setup is still one-shot, as only a single example per class from $L'$ is used.

### 4.1.1 Omniglot

Omniglot [14] consists of 1623 characters from 50 different alphabets. Each of these was hand drawn by 20 different people. The large number of classes (characters) with relatively few data per class (20), makes this an ideal data set for testing small-scale one-shot classification. The $N$-way Omniglot task setup is as follows: pick $N$ unseen character classes, independent of alphabet, as $L$. Provide the model with one drawing of each of the $N$ characters as $S \sim L$ and a batch $B \sim L$. Following [21], we augmented the data set with random rotations by multiples of 90 degrees and used 1200 characters for training, and the remaining character classes for evaluation.

We used a simple yet powerful CNN as the embedding function – consisting of a stack of modules, each of which is a $3 \times 3$ convolution with 64 filters followed by batch normalization [10], a Relu non-linearity and $2 \times 2$ max-pooling. We resized all the images to $28 \times 28$ so that, when we stack 4 modules, the resulting feature map is $1 \times 1 \times 64$, resulting in our embedding function $f(x)$. A fully connected layer followed by a softmax non-linearity is used to define the Baseline Classifier.

Results comparing the baselines to our model on Omniglot are shown in Table 1. For both 1-shot and 5-shot, 5-way and 20-way, our model outperforms the baselines. There are no major surprises in these results: using more examples for k-shot classification helps all models, and 5-way is easier than 20-way. We note that the Baseline Classifier improves a bit when fine tuning on $S'$, and using cosine distance versus training a small softmax from the small training set (thus requiring fine tuning) also performs well. Siamese nets fare well versus our Matching Nets when using 5 examples per class, but their performance degrades rapidly in one-shot. Fully Conditional Embeddings (FCE) did not seem to help much and were left out of the table due to space constraints.

Like the authors in [11], we also test our method trained on Omniglot on a completely disjoint task – one-shot, 10 way MNIST classification. The Baseline Classifier does about 63% accuracy whereas (as reported in their paper) the Siamese Nets do 70%. Our model achieves 72%.

### 4.1.2 ImageNet

Our experiments followed the same setup as Omniglot for testing, but we considered a *rand* and a *dogs* (harder) setup. In the *rand* setup, we removed 118 labels at random from the training set, then tested only on these 118 classes (which we denote as $L_{rand}$). For the *dogs* setup, we removed all classes in ImageNet descended from dogs (totalling 118) and trained on all non-dog classes, then tested on dog classes ($L_{dogs}$). ImageNet is a notoriously large data set which can be quite a feat of engineering and infrastructure to run experiments upon it, requiring many resources. Thus, as well as using the full ImageNet data set, we devised a new data set – *mini*ImageNet – consisting of $60,000$ colour images of size $84 \times 84$ with 100 classes, each having 600 examples. This dataset is more complex than CIFAR10 [12], but fits in memory on modern machines, making it very convenient for rapid prototyping and experimentation. We used 80 classes for training and tested on the remaining 20 classes. In total, thus, we have *rand*ImageNet, *dogs*ImageNet, and *mini*ImageNet.

The results of the *mini*ImageNet experiments are shown in Table 2. As with Omniglot, Matching Networks outperform the baselines. However, *mini*ImageNet is a much harder task than Omniglot which allowed us to evaluate Full Contextual Embeddings (FCE) sensibly (on Omniglot it made no difference). As we an see, FCE improves the performance of Matching Networks, with and without fine tuning, typically improving performance by around two percentage points.

Next we turned to experiments based upon full size, full scale ImageNet. Our baseline classifier for this data set was Inception [25] trained to classify on all classes except those in the test set of classes (for *rand*ImageNet) or those concerning dogs (for *dogs*ImageNet). We also compared to features from an Inception Oracle classifier trained on all classes in ImageNet, as an upper bound. Our Baseline Classifier is one of the strongest published ImageNet models at 79% top-1 accuracy on the standard ImageNet validation set. Instead of training Matching Networks from scratch on these large tasks, we initialised their feature extractors $f$ and $g$ with the parameters from the Inception classifier (pretrained on the appropriate subset of the data) and then further trained the resulting network on random 5-way

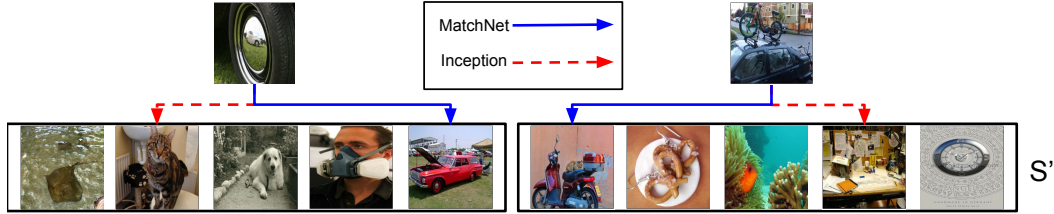

Figure 2: Example of two 5-way problem instance on ImageNet. The images in the set $S'$ contain classes never seen during training. Our model makes far less mistakes than the Inception baseline.

Table 2: Results on *mini*ImageNet.

| Model | Matching Fn | Fine Tune | 5-way Acc 1-shot | 5-way Acc 5-shot |
|---|---|---|---|---|
| PIXELS | Cosine | N | 23.0% | 26.6% |
| BASELINE CLASSIFIER | Cosine | N | 36.6% | 46.0% |
| BASELINE CLASSIFIER | Cosine | Y | 36.2% | 52.2% |
| BASELINE CLASSIFIER | Softmax | Y | 38.4% | 51.2% |
| MATCHING NETS (OURS) | Cosine | N | 41.2% | 56.2% |
| MATCHING NETS (OURS) | Cosine | Y | 42.4% | 58.0% |
| MATCHING NETS (OURS) | Cosine (FCE) | N | 44.2% | 57.0% |
| MATCHING NETS (OURS) | Cosine (FCE) | Y | **46.6%** | **60.0%** |

1-shot tasks from the *training* data set, incorporating Full Context Embeddings and our Matching Networks and training strategy.

The results of the *rand*ImageNet and *dogs*ImageNet experiments are shown in Table 3. The Inception Oracle (trained on all classes) performs almost perfectly when restricted to 5 classes only, which is not too surprising given its impressive top-1 accuracy. When trained solely on $\neq L_{rand}$, Matching Nets improve upon Inception by almost $6\%$ when tested on $L_{rand}$, halving the errors. Figure 2 shows two instances of 5-way one-shot learning, where Inception fails. Looking at all the errors, Inception appears to sometimes prefer an image above all others (these images tend to be cluttered like the example in the second column, or more constant in color). Matching Nets, on the other hand, manage to recover from these outliers that sometimes appear in the support set $S'$.

Matching Nets manage to improve upon Inception on the complementary subset $\neq L_{dogs}$ (although this setup is not one-shot, as the feature extraction has been trained on these labels). However, on the much more challenging $L_{dogs}$ subset, our model degrades by $1\%$. We hypothesize this to the fact that the sampled set during training, $S$, comes from a random distribution of labels (from $\neq L_{dogs}$), whereas the testing support set $S'$ from $L_{dogs}$ contains similar classes, more akin to fine grained classification. Thus, we believe that if we adapted our training strategy to samples $S$ from fine grained sets of labels instead of sampling uniformly from the leafs of the ImageNet class tree, improvements could be attained. We leave this as future work.

Table 3: Results on full ImageNet on *rand* and *dogs* one-shot tasks. Note that $\neq L_{rand}$ and $\neq L_{dogs}$ are sets of classes which are seen during training, but are provided for completeness.

| Model | Matching Fn | Fine Tune | ImageNet 5-way 1-shot Acc $L_{rand}$ | $\neq L_{rand}$ | $L_{dogs}$ | $\neq L_{dogs}$ |
|---|---|---|---|---|---|---|
| PIXELS | Cosine | N | 42.0% | 42.8% | 41.4% | 43.0% |
| INCEPTION CLASSIFIER | Cosine | N | 87.6% | 92.6% | **59.8%** | 90.0% |
| MATCHING NETS (OURS) | Cosine (FCE) | N | **93.2%** | **97.0%** | 58.8% | **96.4%** |
| INCEPTION ORACLE | Softmax (Full) | Y (Full) | $\approx 99\%$ | $\approx 99\%$ | $\approx 99\%$ | $\approx 99\%$ |

### 4.1.3 One-Shot Language Modeling

We also introduce a new one-shot language task which is analogous to those examined for images. The task is as follows: given a query sentence with a missing word in it, and a support *set* of sentences which each have a missing word and a corresponding 1-hot label, choose the label from the support set that best matches the query sentence. Here we show a single example, though note that the words on the right are not provided and the labels for the set are given as 1-hot-of-5 vectors.

```
1.  an experimental vaccine can alter the immune response of people infected with the aids virus a      prominent
<blank_token> u.s.  scientist said.
2.  the show one of five new nbc <blank_token> is the second casualty of the three networks so far      series
this fall.
3.  however since eastern first filed for chapter N protection march N it has consistently promised     dollar
to pay creditors N cents on the <blank_token>.
4.  we had a lot of people who threw in the <blank_token> today said <unk> ellis a partner in            towel
benjamin jacobson & sons a specialist in trading ual stock on the big board.
5.  it's not easy to roll out something that <blank_token> and make it pay mr. jacob says.               comprehensive
```
```
Query:  in late new york trading yesterday the <blank_token> was quoted at N marks down from N          dollar
marks late friday and at N yen down from N yen late friday.
```

Sentences were taken from the Penn Treebank dataset [15]. On each trial, we make sure that the set and batch are populated with sentences that are non-overlapping. This means that we do not use words with very low frequency counts; e.g. if there is only a single sentence for a given word we do not use this data since the sentence would need to be in both the set and the batch. As with the image tasks, each trial consisted of a 5 way choice between the classes available in the set. We used a batch size of 20 throughout the sentence matching task and varied the set size across k=1,2,3. We ensured that the same number of sentences were available for each class in the set. We split the words into a randomly sampled 9000 for training and 1000 for testing, and we used the standard test set to report results. Thus, neither the words nor the sentences used during test time had been seen during training.

We compared our one-shot matching model to an oracle LSTM language model (LSTM-LM) [30] trained on all the words. In this setup, the LSTM has an unfair advantage as it is not doing one-shot learning but seeing all the data – thus, this should be taken as an upper bound. To do so, we examined a similar setup wherein a sentence was presented to the model with a single word filled in with 5 different possible words (including the correct answer). For each of these 5 sentences the model gave a log-likelihood and the max of these was taken to be the choice of the model.

As with the other 5 way choice tasks, chance performance on this task was 20%. The LSTM language model oracle achieved an upper bound of 72.8% accuracy on the test set. Matching Networks with a simple encoding model achieve 32.4%, 36.1%, 38.2% accuracy on the task with $k = 1, 2, 3$ examples in the set, respectively. Future work should explore combining parametric models such as an LSTM-LM with non-parametric components such as the Matching Networks explored here.

Two related tasks are the CNN QA test of entity prediction from news articles [5], and the Children's Book Test (CBT) [6]. In the CBT for example, a sequence of sentences from a book are provided as context. In the final sentence one of the words, which has appeared in a previous sentence, is missing. The task is to choose the correct word to fill in this blank from a small set of words given as possible answers, all of which occur in the preceding sentences. In our sentence matching task the sentences provided in the set are randomly drawn from the PTB corpus and are related to the sentences in the query batch only by the fact that they share a word. In contrast to CBT and CNN dataset, they provide only a generic rather than specific sequential context.

## 5 Conclusion

In this paper we introduced Matching Networks, a new neural architecture that, by way of its corresponding training regime, is capable of state-of-the-art performance on a variety of one-shot classification tasks. There are a few key insights in this work. Firstly, one-shot learning is much easier if you train the network to do one-shot learning. Secondly, non-parametric structures in a neural network make it easier for networks to remember and adapt to new training sets in the same tasks. Combining these observations together yields Matching Networks. Further, we have defined new one-shot tasks on ImageNet, a reduced version of ImageNet (for rapid experimentation), and a language modeling task. An obvious drawback of our model is the fact that, as the support set $S$ grows in size, the computation for each gradient update becomes more expensive. Although there are sparse and sampling-based methods to alleviate this, much of our future efforts will concentrate around this limitation. Further, as exemplified in the ImageNet dogs subtask, when the label distribution has obvious biases (such as being fine grained), our model suffers. We feel this is an area with exciting challenges which we hope to keep improving in future work.

## Acknowledgements

We would like to thank Nal Kalchbrenner for brainstorming around the design of the function $g$, and Sander Dieleman and Sergio Guadarrama for their help setting up ImageNet. We would also like thank Simon Osindero for useful discussions around the tasks discussed in this paper, and Theophane Weber and Remi Munos for following some early developments. Karen Simonyan and David Silver helped with the manuscript, as well as many at Google DeepMind. Thanks also to Geoff Hinton and Alex Toshev for discussions about our results, and to the anonymous reviewers for great suggestions.

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
