[Reviews · NeurIPS 2016]

Reviewer 1

Summary

In this paper the authors phrase one-shot learning as an instance of a “set-to-set” neural network, that has a very similar structure as various recently published “sequence-to-sequence” models; or memory- / pointer networks. The idea and proposed architecture is simple and compelling. The paper also contains an extensive experimental part where multiple new one-show problems are introduced.

Qualitative Assessment

The basic idea behind this paper is simple and seems compelling. Until about line 110, the authors do a good job motivating and explaining their model. The paragraph about LSTM equipped set-embeddings (lines ~116 to 123) is not very detailed though. I can imagine many different architectures that fit with the description in this paragraph. Furthermore, I could not find detailed descriptions of the models and hyperparameters used for the experiments -- neither in the experimental section nor in the (non-existing) supplemental material. From this perspective it seems hard to independently reproduce the presented results. Besides of introducing a new model, the paper also introduces various new one-shot learning tasks. The tasks seem reasonable hard and the authors evaluated various baseline models against these. In summary: this paper introduces a simple and interesting method to train one-shot learning classifiers. The experimental section is extensive and the proposed model seems to outperform previous state-of-the-art methods considerably; but given that many of the tasks are new, it is very hard for me to assess the actual (empirical) progress archived by the method.

Confidence in this Review

2-Confident (read it all; understood it all reasonably well)


Reviewer 2

Summary

The article describes a one-shot learning model called matching network. The basic idea of the model is to compute the output category of one given test example by computing a kind of ‘similarity’ with all the training examples that are stored in a memory. This similarity (denoted a(x,x_i)) is based on two representation functions f and g which goals are to encode both the new example, but also the examples stored in memory. The proposed model is thus very close to siamese networks, but using an asymmetric architectures since g and f can be different functions. The main originality is contained in two particular aspects: first, instead of computing the g function on each training example, the model proposes a sequential model (bidirectional LSTM) to learn to encode the training examples based on the previously seen examples. Second, the set of examples is also used to learn to encode any incoming datapoint. These two components will be learned over a set of one-shot learning tasks, resulting in a ‘learning to learn’ problem. Experiments are made on different images datasets and one language dataset. The proposed approach is compared to baseline models, MANN and siamese nets (not on all the datasets…) and shows interesting results

Qualitative Assessment

Comments: the idea contained in the paper is not a very big contribution to the field but still remains interesting, and the experiments on different datasets provide important information concerning the behavior of the model. But the paper has many different problems that make it difficult to understand. The first problem is in the structure of the paper since the problem formulation is in fact only given in Section 2.2 while I think it is important to well define the one-shot learning problem at the beginning of the paper. It would make the article easier to read and to follow. Sections 2.1 and 2.1.1 are well written but could include more precise definitions. For example, in Equation 1, y_i is not defined : is it a real number, a one hot vector (I suppose it is a one hot vector). The core of the contributions in Section 2.1.2 is very difficult to well understand. It is not clear how the bidirectional LSTM is used, but the more difficult part concerns the way f(\hat(x)) is defined. The authors could clearly give more details since this section is in fact the main contribution of the article. Written as it is, it makes the model difficult to understand, and almost impossible to reproduce. At last, if these sections can be understood by researchers very familiar to deep learning, it will not be the case for ML researchers used to read more ‘classical’ papers while the principle of the paper is more general than deep learning and can be for example understood from a metric learning point of view. I clearly advise the authors to improve the way the model is described. Concerning the experiments, it is really interesting to see that this type of approach improves the quality of the system w.r.t Siamese networks. But the Siamese net results are only provided for omniglot and not for imagenet which is a little bit strange. The language model part also does not contain any table/figure showing a summary of the results which make the discourse difficult to follow in this section. To conclude, if the article provides a simple but interesting idea for one-shot learning, the way it is written must be drastically improved to allow a publication in a top conference.

Confidence in this Review

3-Expert (read the paper in detail, know the area, quite certain of my opinion)


Reviewer 3

Summary

The paper proposes a non-parametric method for one-shot learning where the weight (or, distance metric) between the test item and its neighbors (one-shot set) can be learnt by back-propagation. For this, it also proposes a meta-learning training strategy where a mini-batch to mini-batch learning is performed. The proposed method is tested on classification tasks with three different datasets. The proposed method seems to work reasonably well compared to the results of previous one-shot learning methods.

Qualitative Assessment

Note that after reading the submitted version, I have also found and read the arXiv version. My comments are based on the arXiv version unless mentioning the source explicitly. Overall, I enjoyed the paper and the performance is interesting. Below are my comments: - It seems good to put the model description in the Appendix to the main description of the paper. Without it, I felt the exposition a bit abstractive and not so clear. Also, I'm wondering the meaning of eqn 4. in the Appendix. - The effect of ordering for items in S. It seems that the FCE for g(x) requires to order the elements in S. What kind of criteria did you use for the ordering and how sensitive the performance by different ordering schemes. - In lines around 80-82 (in the submission version), it says "Unlike other attentional memory mechcanisms [2], (1) is non-parameteric in nature: as the support set size grows, so does the memory used". -> I'm not so sure about this. Doesn't the memory size also change depending on e.g., the input sentence length (e.g., in NMT)? - I would add more detailed description on the convolutional siamese networks because it is one of the main algorithms compared in the experiments. - I would still like to see the actual numbers for the FCE version in the Table 1. although the authors mention that "it did not seem to help much" and some analysis on why the FCE was not helpful for omniglot is required. - For the language modeling experiments, I would like to see at least one other one-shot learning method in the comparison. Minor comments - Some explanation is required in the caption of Fig. 1 which is also too large. - More informative notation would be helpful. E.g. in Eqn. (1) I guess the label y_i will be the softmax output vector for classification. Current notation makes it look like weighted average of the integer class labels. - Also, notation, sometimes use Equantion X and other times Eq. X. etc. - Ref. for meta-learning in line 139.

Confidence in this Review

2-Confident (read it all; understood it all reasonably well)


Reviewer 4

Summary

This paper presents Matching Nets, a one-shot learning framework that adopts advances in attention and memory that enable rapid learning. It maps a small labelled support set to a classifier, where the mapping is parameterized by a neural network. They also define one-shot tasks that can benchmark other approaches on ImageNet and small scale language modeling.

Qualitative Assessment

Matching Networks aims to incorporate the best characteristics from both parametric and non-parametric models, providing rapid acquisition of new examples as well as generalization from common examples. They cast the problem of one-shot learning within the set-to-set framework, which seems an extension of their previous work. The main novelty of MN, as claimed in the paper, is reinterpreting a well studied framework to do one-shot learning. However, it is kind of hard to claim this as novelty. The difference with simple KDE/NN is that MN adopts the parameterized mapping as feature extractor. MN tries to train two embeddings (CNN/LSTM) to minimize the classification error between support set and unseen examples. Given the limits mentioned in the conclusion, this paper seems like a preliminary work on framework/task definition, which does not solve real challenges, e.g. speed/scalability issues with large training data. Some minor problems: 1. Figure 1 is not referred in the text and needs better explanation. For example, what is the definition of g_theta and f_theta? They are not mentioned until sec 2.1.1. 2. The meaning of “switching the task from minibatch to minibatch” in line 35 is unclear . 3. The organization of sec 4 is a little wired, should section of 4.1.3 be 4.2?

Confidence in this Review

1-Less confident (might not have understood significant parts)


Reviewer 5

Summary

Deep supervised networks do not typically adapt rapidly to new concepts from sparse data. This paper alleviates this problem by proposing a neural network based model with a non-parametric structure. One of the main high-level insights in this paper is that for one-shot learning problems, train and test both on one-shot like settings. Also the use of non-parametric ideas in deep learning is an interesting way to deal with catastrophic forgetting and to robustly adapt to fast changes in the concept space. For a new test item, the model predicts the label as the weighted linear sum of labels in the training support set S. This weighting is computed by taking the cosine distance between features of the test item and features of items in S. Two separate neural networks are used for the test items (g) and items in S (f) respectively. Both g and f are recurrent neural networks which embed data-points in the context of S (which enables data-dependent embeddings). I really like this paper as it proposes a very simple and elegant approach to one shot learning by combining neural networks with non-parametric structures. The contributions can be summarized as: (1) To do one shot learning during test, train the network to do one-shot learning (2) Incorporate non-parametric structure into neural networks via an attention mechanism (3) Authors demonstrate state of the art classification accuracy on Omniglot

Qualitative Assessment

- Are the omniglot results on the same train-test split as reported in Lake et al ? A head on comparison with this paper would be interesting and important to see. - What happens with the quantitative results when the embedding functions f and g are same architectures/networks. - One of the most unsatisfying things about this model is that it won't perform well when the training and test tasks come from a significantly different distribution. The authors do present some analysis in 4.2.1 that this limitation might be causing performance degradation in L_dogs. I would have liked to see more of such test cases. - For reproducibility, authors should list network architecture details for each setup Overall I really like the direction of adapting neural nets for one-shot learning and this paper makes meaningful scientific contributions towards it.

Confidence in this Review

3-Expert (read the paper in detail, know the area, quite certain of my opinion)


Reviewer 6

Summary

The paper presents a non-parametric approach to one shot learning. Novelties claimed are two-fold. First, the network itself utilizes ideas from attention and memory and second the training procedure keeps switching the task every minibatch. The model and training strategy is explained well on most parts, however, there are a few important that currently seem unanswered. Experimental results show that the model is in fact able to learn the underlying structure of the training classes and is able to generalize well enough to novel classes.

Qualitative Assessment

Major comments: 1) The model utilizes a bidirectional LSTM to encode a sample along with the context of an entire support set S. In line 120, the paper mentions that this allows the model to "ignore" some elements in the support set S. Further, "depth" is added to the computation. This reviewer feels it would be helpful to have more discussion on both these points, further motivating the use of sequence modelling. Along the same lines, line 122 could be extended to include more discussion, strengthening the connections to the "learning to learn" paradigm. 2) Although the network performance is good on the tasks that the paper defines, it is unclear how "similar" networks would perform. For examples, exploratory experiments could be performed to investigate how important is the application of sequence modelling to the task. How do other models that take in a set along with attention but do not utilize sequence modelling perform ? Investigative studies prove to be more useful to the overall understanding of the phenomenon that drives the best performing model. Perhaps the authors can simply add in some discussion into what alternate (but valid) models can be used and hypothesize how would they behave, although ideally, atleast one exploratory experiment could be reported. These were the main reasons this reviewer believes that the the clarity/presentation of the paper is sub-standard for NIPS. The paper is clear on many significant parts, but given the importance of the problem addressed, and the novelties, more clarity/understanding of the overall model (alternate valid models) might be required. Minor comments: 1) Fig. 1 caption could give more details on the algorithm. It helps to have an informative figure near the beginning of the paper.

Confidence in this Review

2-Confident (read it all; understood it all reasonably well)